# Examining the safety of menstrual cups among rural primary school girls in western Kenya: observational studies nested in a randomised controlled feasibility study

Jane Juma,[1] Elizabeth Nyothach,[1] Kayla F Laserson,[2] Clifford Oduor,[1] Lilian Arita,[1] Caroline Ouma,[1] Kelvin Oruko,[1] Jackton Omoto,[3] Linda Mason,[4] Kelly T Alexander,[4] Barry Fields,[2] Clayton Onyango,[2] Penelope A Phillips-Howard[5]

► Prepublication history and additional material are available. To view these files please visit the journal online (http://dx.doi.org/ 10.1136/ bmjopen-2016-015429).

[1]Center for Global Health, Kenya Medical Research Institute, Kisumu, Kenya
[2]Centers for Disease Control and Prevention, Atlanta, Georgia, USA
[3]Siaya District Hospital, Ministry of Health, Siaya, Kenya
[4]Liverpool School of Tropical Medicine, Liverpool, UK
[5]Department of Clinical Sciences, Liverpool School of Tropical Medicine, Liverpool, UK

**Correspondence to**
Dr Penelope A Phillips-Howard; Penelope.Phillips-Howard@ lstmed.ac.uk

## ABSTRACT

**Objective** Examine the safety of menstrual cups against sanitary pads and usual practice in Kenyan schoolgirls.

**Design** Observational studies nested in a cluster randomised controlled feasibility study.

**Setting** 30 primary schools in a health and demographic surveillance system in rural western Kenya.

**Participants** Menstruating primary schoolgirls aged 14–16 years participating in a menstrual feasibility study.

**Interventions** Insertable menstrual cup, monthly sanitary pads or 'usual practice' (controls).

**Outcome measures** *Staphylococcus aureus* vaginal colonization, *Escherichia coli* growth on sampled used cups, toxic shock syndrome or other adverse health outcomes.

**Results** Among 604 eligible girls tested, no adverse event or TSS was detected over a median 10.9 months follow-up. *S. aureus* prevalence was 10.8%, with no significant difference over intervention time or between groups. Of 65 *S. aureus* positives at first test, 49 girls were retested and 10 (20.4%) remained positive. Of these, two (20%) sample isolates tested positive for toxic shock syndrome toxin-1; both girls were provided pads and were clinically healthy. Seven per cent of cups required replacements for loss, damage, dropping in a latrine or a poor fit. Of 30 used cups processed for *E. coli* growth, 13 (37.1%, 95% CI 21.1% to 53.1%) had growth. *E. coli* growth was greatest in newer compared with established users (53%vs22.2%, p=0.12).

**Conclusions** Among this feasibility sample, no evidence emerged to indicate menstrual cups are hazardous or cause health harms among rural Kenyan schoolgirls, but large-scale trials and post-marketing surveillance should continue to evaluate cup safety.

## Strengths and limitations of this study

► In the small sample of girls followed, there was no evidence of health harms.
► Evaluation of the safety of menstrual products, including laboratory investigations, was feasible among adolescent schoolgirls in a low/middle-income country (LMIC) setting.
► Logistical limitations prevented 'before' and 'after' prevalence surveys.
► To minimise the possible health risks, girls were trained on how to use and clean menstrual cups, and girls in all arms were provided soap for handwashing, with follow-up by nurses, creating improved hygiene circumstances for cup use in this LMIC setting.

these females may manage menstruation with non-absorbent, unhygienic and uncomfortable materials.[6–9] Studies in southern Asia and Africa report that such items are associated with genital infections although these are seldom clinically verified,[10–12] preventing the understanding of women's needs to minimise such risks.[2 6–8 13] Meanwhile, a growing body of pilot studies have embarked on testing the value of menstrual hygiene products, such as sanitary pads[14–16] and menstrual cups,[17–22] for girls and women in resource-poor settings.

Although menstrual cups have not been associated with an increased risk of reproductive tract and urogenital infections in women in high-income countries,[23–28] research on the safety of menstrual cups among girls[17 19 21] and women[18 20] in LMIC has relied on self-reported information with no clinical or laboratory confirmatory studies. There is a concern that an insertable menstrual item may increase the risk of

## INTRODUCTION

The inadequate management of adolescent girls' menstruation in low/middle-income countries (LMIC) has recently emerged as an important priority for international action.[1–5] Due to logistical and cost barriers,

infections, particularly *Staphylococcus aureus*, leading to menstrual toxic shock syndrome (mTSS).[29] Tampons are linked to mTSS in women of reproductive age. Surveillance data for the period 1979–1996 indicate that 5296 cases were reported in women in the USA using highly absorbent tampons.[30] The tampons were found to have been associated with vaginal microtrauma arising from the high absorbency.[27 30–33] Menstrual cups which collect menstrual blood, however, are non-absorptive and do not disrupt the vaginal epithelium.[27 28] Furthermore, among women using female barrier methods, which similarly uses medical grade silicone or latex products, mTSS is very low (~2.25 cases per 100 000 users per year).[34] Nevertheless, concern remains about any vaginal intrusion,[29 35 36] particularly among girls,[37] with poor water, sanitation and hygiene (WASH) facilities.[38] Further laboratory and field-based studies are, thus, needed to clarify risks associated with menstrual products to better define the cost-benefit of subsidised provision for girls in LMIC. This paper describes the exploration of cup safety during a randomised controlled pilot feasibility study among adolescent schoolgirls in rural Kenya.[39]

## METHODS

### Study site and population

The study site is within the health and demographic surveillance system (HDSS) of the Kenya Medical Research Institute (KEMRI) and CDC research station in Siaya County, a rural district in the former Nyanza Province, in western Kenya.[40] The population of the HDSS site approximates 230 000 individuals in a ~700 km$^2$ area, with adolescent girls aged 15–19 years comprising ~11% of the female population.[40] The area is served by 1 district hospital providing tertiary care and 10 local health clinics within the HDSS area. A second district hospital is sited in Kisumu, 40 km from the study site. The KEMRI and CDC collaborative research station has on-site laboratories certified for clinical trials and quality control procedures.

### Menstrual Solutions study

The Menstrual Solution study was a cluster randomised controlled 3-arm 'proof of concept' feasibility study conducted in Gem, a subarea within the HDSS.[39] Of 71 primary schools in Gem, 62 agreed to participate in a baseline water sanitation and hygiene (WASH) assessment. Of these, 30 reached the pre-defined WASH threshold (presence of water in school on the day of WASH visit, availability of separate latrine bank for girls and a pupil:latrine ratio of 70:1 or less) described in detail elsewhere.[38] All girls in the 30 study schools were eligible if they were aged 14–16 years, had experienced 3 menses, were resident in the HDSS for at least 4 months and attended a study school (figure 1).[39] A sample of 185 girls/arm was estimated to offer 5% precision for the primary outcome, (school dropout) if this occurred in 15% of the control arm.[39] Recruitment of 250 girls (10 schools, average of 25 girls per school) was scheduled to allow for a design effect

of 1.25% and 7.5% loss to follow-up. Girls were enrolled from August 15, 2012 to August 27, 2013 and followed until November 21, 2013, with a median (IQR) follow-up time of 10.9 (6.1–12.5) months. Further details of the overall study methods are published elsewhere.[39]

### Menstrual products and hygiene

Girls in the menstrual cup group were provided with one menstrual cup (Mooncup), size B for nulliparous women or size A for those who had given birth (figure 2). This brand was selected because it has been tested internationally,[26 41] registered by the US Food and Drug Administration and by the Kenyan Pharmacy and Poisons Board for pilot testing among schoolgirls in Nairobi.[19] Cups are made of high-grade medical silicone with material continuous without edges.[25] When inserted into the vagina, it collects ~30 mL of menstrual blood, lasting 4–8 hours before emptying is required, according to the manufacturer. Girls in the cup group were given instruction on how to insert, remove and clean the cup. Girls in the sanitary pad arm were each given two packs (total 16 pads) monthly of Always, a brand available in Kenya. Girls in the usual practice group continued using traditional materials, such as cloths, bedding or paper[2 22] or sanitary pads. All participating girls, regardless of study arm, received a lesson on menstrual hygiene by study nurses, including hand-washing and how to wipe after defecation, and provision of bar soap for hand hygiene throughout the study. Schools separately received soap detergent to support pupils' hand-washing in school.

### Safety monitoring

Safety monitoring components comprised routine nurse-based screening, population-based monitoring (figure 3) and clinical evaluation of infection with laboratory confirmation.

### Nurse screening and population-based monitoring

Following a meeting with the girls and parents, each family was provided with information leaflets which included signs and symptoms of mTSS, where they could seek emergency care and contact information. Possible mTSS was monitored through school and community pathways. Study nurses screened each participant routinely twice per term, asking girls about comfort of the product, use and any health concerns including questioning of mTSS signs and symptoms. This was supplemented by nurse visits to their designated target schools one to two times a week, allowing examination of any participant complaining of any signs or symptoms that could be mTSS or another infection or harm. A focal point teacher designated by girls in each school was provided credit for her phone to communicate with nurses between visits, if necessary. Field staff were informed of all girls participating in the study who resided in their designated village. All participants and their families were provided contact details of the field staff in their village, in case they wished to contact them in the event of a febrile episode or any

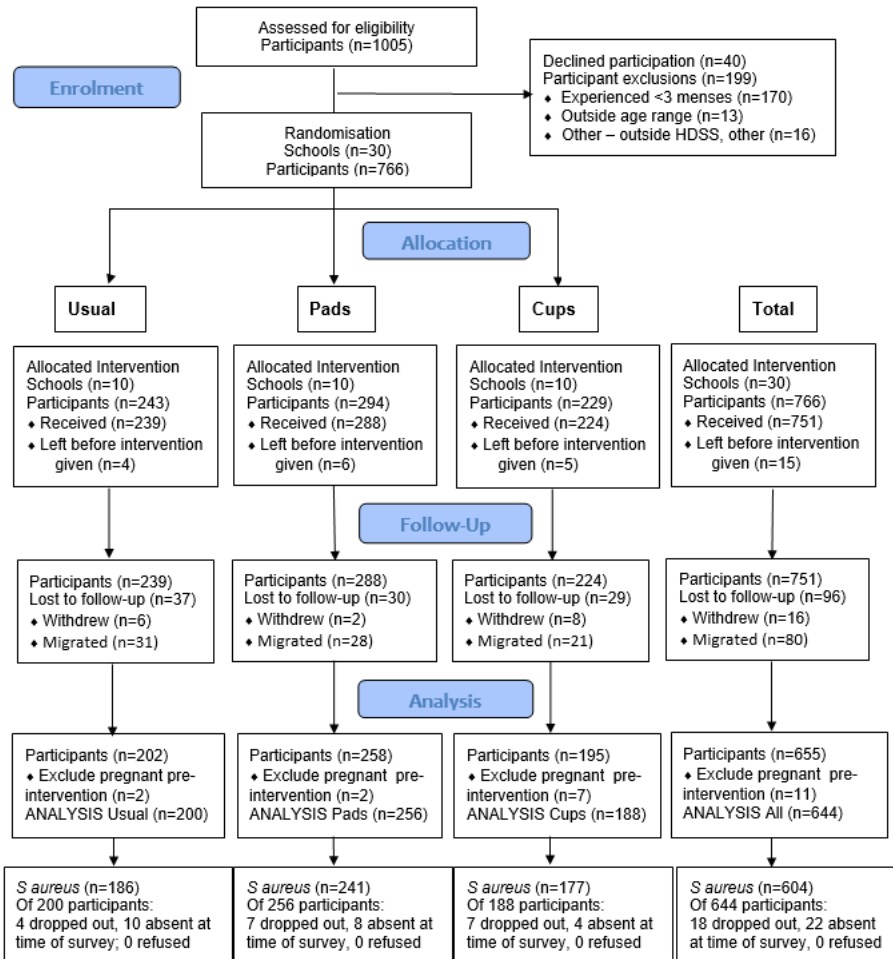

**Figure 1** Participants' flow diagram for Menstrual Solution study and *Staphylococcus aureus* survey.

other symptoms that could have been mTSS. The field staff were provided contact information for the study nurses and research team, and could support immediate evacuation to tertiary care facilities, if required.

The hospitals and health facilities in the study area were provided with leaflets on clinical signs and symptoms of mTSS. Consultant obstetrician and gynaecologists in tertiary care facilities were involved with the study and were prepared to accept any mTSS referral cases during the study duration. At the end of the study, nurses visited all health facilities in the study area to recheck registry records as a precaution against any case missed by our monitoring procedures. The study data manager reviewed the HDSS database for deaths of any study participants on a monthly basis, and completed a summary check at study end to enumerate and confirm if any deaths had occurred or been missed from the monthly surveillance checks.

### Vaginal swabbing to evaluate the prevalence of *S. aureus*

Assenting girls in all groups were invited to have a vaginal swab to examine the prevalence of *S. aureus* colonisation between January and September 2013. After piloting the procedure, each nurse facilitated the *S. aureus* study in her schools, training the girls on the self-swabbing procedure.[42] The self-collected vaginal swab (BBL Culture swab, COPLAN for Becton Dickinson) was conducted by girls in a school bathroom or private room. Participants were taught to insert the swab into the endocervical canal and stop when the tip was no longer visible. As instructed, the girls would then rotate the swab three to five times inside the vagina, withdraw it but avoid contact with vaginal surfaces, and put the swab in the tube (containing Ames

**Figure 2** Menstrual cup distributed to girls in cup-allocated schools.

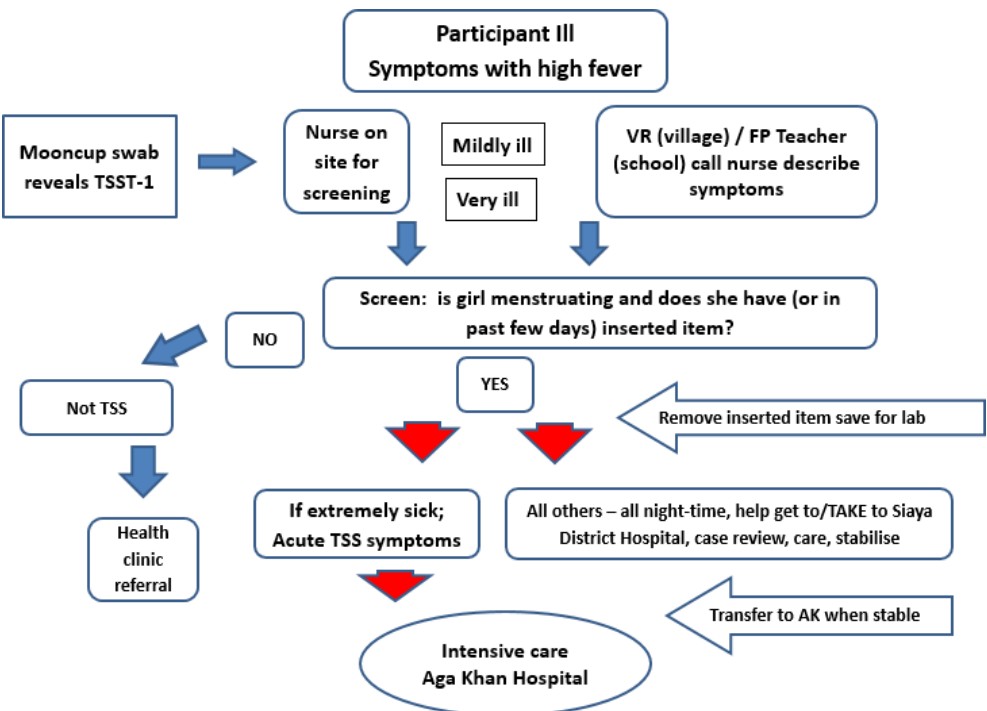

**Figure 3** Flow diagram for action of suspected menstrual toxic shock syndrome event. AK, Aga Khan; FP, focal point; TSST, toxic shock syndrome toxin;VR, village recorder (field staff).

transport media with agar), break the handle of the swab and close the tube tightly. Swabs were not taken during menstruation.[29 42–44] The girls gave the vaginal swab to the nurse who checked whether it was moist/discoloured before placing it in a phial containing Amies transport medium with agar (Becton Dickinson Microbiology Systems). Each phial was labelled individually with the girls' unique study code, and placed in a cool box packed with frozen ice packs, and shipped to the KEMRI laboratories for isolation and phenotypic identification of *S. aureus* and TSST-1 production. Swabs were transported on the same day, within 6 hours of collection, to the KEMRI laboratory, and stored at −70°C to −80°C before testing.

### Swab analysis for *S. aureus*

Laboratory staff were masked to girls' allocated menstrual product. Each vaginal swab was processed as per protocol.[27 29] Swabs were streaked for isolation onto a mannitol salt agar plate and on tryptic soy agar with 5% sheep blood. All plates were incubated between 36°C and 37°C for 24 hours in air. After incubation, colony types were visualised for characteristic morphology of *S. aureus*. Colonies were enumerated, isolated on plates containing tryptic soy agar, and incubated for 24 hours between 36°C and 37°C. Gram staining, a catalase test, and a slide and tube coagulase test were performed to phenotypically identify *S. aureus*.

### Detection of TSST-1 among *S. aureus*-positive girls

This required a second (repeat) positive swab taken from girls found positive in the main prevalence survey. Repeat swabs from *S. aureus*-positive girls were processed as above,

and examined again for the presence of *S. aureus*. After positive identification as *S. aureus* via culture (required for toxic shock syndrome toxin-1 (TSST-1) testing), isolates were placed in sterile (16 mm by 150 mm) glass tubes containing 5 mL of brain heart infusion broth (Becton Dickinson), and tubes were mixed end over end for 18–24 hours at 36°C–37°C in a shaking incubator. Samples were pelleted by centrifugation (900 $g$ for 20 min at 4°C) and the supernatant placed into microcentrifuge tubes. The Toxic Shock Test-Reverse Passive latex Antigen (TST-RPLA) agglutination test for the detection of TSST-1 in the culture fluid supernatant of the cultured *S. aureus* was tested using the staphylococcal test kit (TD0940A).

### Menstrual cup screening for *Escherichia coli* contamination

Data from all girls receiving cups were assessed and 1:4 cups were randomly selected stratified by duration of provision, excluding girls who had received a replacement. Randomly selected participants were traced and asked if they were willing to swap their existing cup for a new one, to allow laboratory examination of their cup. Each used cup was placed in a separate lock-bag, which was labelled with girls' study code, and transported to the laboratory and tested for *E. coli* growth. Each cup was swabbed using a polyester tipped swab moistened in normal saline and inoculated into both MacConkey (MAC) agar and onto blood agar and incubated for 18–24 hours at 37°C. After incubation, colony types were visualised for characteristic morphology of *E. coli* from the MAC plates and subjected to indole testing. The colonies generated which were indole positive were classified using standard terminology

as suspected *E. coli*.[45] Laboratory outcomes were analysed by study arm, into proportion positive, with 95% confidence limits.

## Data analysis

Participant characteristics gathered through girls' self-completed surveys on notebooks, intervention implementation by date of provision and duration of follow-up by study nurses, and laboratory results were aggregated by participant ID, and prevalence values analysed using SPSS version 21.0. The prevalence of *E. coli* on cups was calculated with 95% CIs. Means and medians were calculated with corresponding SD and the IQR. Significant differences in prevalence and linear trends were tested using $\chi^2$.

## RESULTS

Of 1005 girls in the 30 study schools in eligible classes, 199 (19.8%) were ineligible, 40 (5.0%) girls refused, and 15 (1.8%) migrated before intervention (figure 1). Of the 751 receiving intervention, 11 were pregnant prior to intervention and 96 (12.8%) were lost to follow-up providing 644 girls for outcome evaluation.[39] Of these, 604 contributed towards the population surveyed for *S. aureus*, and 40 girls were not swabbed (18 had dropped out and 22 were absent at the time of survey).

## Participant characteristics

The mean (SD) age of participants at enrolment was 14.6 (0.7) years, and mean age at menarche was 13.6 (0.9) years (table 1). Menses lasted for a mean of 3.8 (1.3) days, with most girls (82.6%) reporting they had ever used pads, but none used tampons or menstrual cups. A quarter reported having ever had sex, there were 4 pregnancies and 12 stated they were married.

## Morbidity and mortality surveillance

No symptoms of mTSS were identified during nurse screening or reported through village recorders. No cases of mTSS were identified or referred to tertiary care facilities. The health clinic records review identified no participants attending health services for febrile episodes or any other symptom of mTSS. HDSS census review monthly and at end study identified no deaths among our study participants. At nurse routine screening, 10 girls (5 pads and 5 cups) reported heavy bleeding, 80% of whom had reported this preintervention. These girls were referred to tertiary care facilities, where the consultant gynaecologist reported no abnormal findings, but provided haematinics.

## *S. aureus* prevalence survey

Of 604 vaginal swabs collected among eligible participants, *S. aureus* was detected in 65 (10.8%) samples (table 2). When stratified by duration of intervention, *S. aureus* prevalence was 13.0% in the first intervention month, with no statistically significant trend between groups (10.5% cups, 13.6% pads, 15.2% controls; $\chi^2$ linear trend=0.34, p=0.56). Prevalence during intervention follow-up, after a median of 4 months (range 2–11 months) was 10.2%, with no significant trend between groups (9.4% cups, 10.7% pads, 10.5% controls; $\chi^2$ linear trend=0.09, p=0.76). There was no significant difference in prevalence between early intervention and during use, overall or within groups (table 2).

## Toxic shock syndrome toxin (TSST)–1

In the first batch of 65 participants with vaginal swabs positive for *S. aureus*, 49 girls were available to collect a

---

**Table 1** Characteristics of study population*

| Characteristics | Statistics/category | Control, % (n=200) | Pads, % (n=256) | Cups, % (n=188) | Total, % (n=644) |
|---|---|---|---|---|---|
| Age in years at enrolment | Mean (SD) | 14.6 (0.7) | 14.5 (0.7) | 14.6 (0.7) | 14.6 (0.7) |
| Age in years at menarche | Mean (SD) | 13.6 (0.8) | 13.7 (0.8) | 13.5 (1.0) | 13.6 (0.9) |
| Number of days of menses | Mean (SD) | 3.7 (1.2) | 3.9 (1.3) | 3.7 (1.5) | 3.8 (1.3) |
| Experience heavy periods | Yes | 41 (20.5%) | 68 (26.6%) | 39 (20.7%) | 148 (23.0%) |
| Experience period cramps | Yes | 129 (64.5%) | 165 (64.5%) | 115 (61.2%) | 409 (63.5%) |
| Ever used pads | Yes | 168 (84.0%) | 198 (77.3%) | 166 (88.3%) | 532 (82.6%) |
| Ever had sex[†] | n | 194 | 249 | 183 | 626 |
| | Yes | 47 (24.2%) | 58 (23.3%) | 58 (31.7%) | 163 (26.0%) |
| Ever been pregnant | n | 194 | 249 | 183 | 626 |
| | Yes | 0 (0%) | 2 (0.8%) | 2 (1.1%) | 4 (0.6%) |
| Report being married | n | 194 | 249 | 183 | 626 |
| | Yes | 3 (1.5%) | 4 (1.6%) | 5 (2.7%) | 12 (1.9%) |
| Duration of follow-up | Median (IQR) | 10.5 (5.6–12.5) | 11.4 (6.7–12.5) | 10.9 (5.0–12.6) | 10.9 (6.1–12.5) |

*Characteristics reported by 644 participants at baseline survey.[39]
†626 of 644 answered questions on sex, pregnancy and marriage; ever had sex includes girls reporting having had sexual intercourse, including those reporting tricked or forced to have sexual intercourse; n - number who answered question.

**Table 2** Prevalence of *Staphylococcus aureus* early and during intervention by study group

| | First month intervention | Greater than 1 month intervention | Total study | $\chi^2$ | p-Value |
|---|---|---|---|---|---|
| Cups | 4/38 (10.5) | 13/139 (9.4) | 17/177 (9.6) | 0.05 | 0.83 |
| Pads | 6/44 (13.6) | 21/197 (10.7) | 27/241 (11.2) | 0.32 | 0.57 |
| Control | 5/33 (15.2) | 16/153 (10.5) | 21/186 (11.3) | 0.6 | 0.44 |
| Total | 15/115 (13.0) | 50/439 (10.2) | 65/604 (10.8) | 0.77 | 0.38 |
| $\chi^2$ linear trend | 0.34 | 0.09 | 0.26 | | |
| p-value | 0.56 | 0.76 | 0.61 | | |

second swab to examine the presence of TSST-1. Of these swabs, 10 (20.4%) yielded *S. aureus*. These second-level swabs were immediately processed to examine the presence of TSST-1 toxin. Of these 10, 2 (20%) tested positive for TSST-1. Neither of these was a girl provided with a menstrual cup; both were in the sanitary pad group. Study participants were followed up, including these two girls, and found to be healthy and asymptomatic.

### Menstrual cup loss or damage
Of 188 girls followed to outcome, 14 (7%) girls required replacement cups due to cup loss (3 participants, including 1 girl twice), damage (two cups: one burned when boiling and one eaten by rats), too small (three cups, replaced with size A due to leaking) or dropped inside the latrine (six cups). Examination of cups during screening revealed only minor abrasions or small damage to tail ends when cut to size.

### *E. coli* growth on used menstrual cups
Of 188 girls provided cups and followed to outcome, 21 dropped out. Of the 167 non-dropouts, a sampling frame was obtained from the nurse's follow-up survey database in the last study quarter. This comprised 134 girls surveyed, with 33 girls missed due to non-attendance.

From this sample of 134 girls, a random selection of 1 in 4 cups (35, 26%) were randomly selected. No girls refused. Duration of provision was used to stratify the 35 cups, with 17 representing new cup users provided cups for less than 6 months and 18 established cup users provided for 6 months or longer (table 3). Five unused cups acted as controls. Of the 40 cups processed, there was no *E. coli* on control cups, whereas 13 of 35 used cups had *E. coli* growth (37.1%, 95% CI 21.1% to 53.1%; table 3). By duration of cup provision, the prevalence of *E. coli* growth generated was greatest in newer users, with growth on 9 of 17 (53%, 95% CI 29.3% to 76.7%) cups compared with 4 of 18 (22.2%, 95% CI 2.9% to 41.1%) cups of established users, a difference of 31% (p=0.12). Examination of *E. coli* growth by girls' age, socioeconomic status, reported ever used pads, age at menarche, duration of bleeding and if periods were reported to be heavy only found an association with heavy periods; 61.5% of girls reporting heavy periods had *E. coli* on cups, compared with 22.7% of those stating they did not have heavy periods (p=0.022).

### DISCUSSION
We observed 10.8% prevalence of *S. aureus* in 14- to 16-year-old girls in this area. This is at the lower range

**Table 3** *Escherichia coli* growth generated on cups over differing time spans

| | | Total cups available* population represented | Cups randomly sampled | Proportion cups from available sample,% | Number with *E. coli* growth | Prevalence,% (95% CI) |
|---|---|---|---|---|---|---|
| New | <2 m use | 14 | 6 | | 3 | 50 (10.0 to 90.0) |
| | 3–5 m use | 51 | 11 | | 6 | 55 (25.6 to 84.4) |
| | All new users (<6 m) | 65 | 17 | 25 | 9 | 53 (29.3 to 76.7) |
| Long term | Established (6–9 m) | 58 | 12 | | 4 | 33.3 (6.6 to 60.0) |
| | Longer term (9 m>) | 11 | 6 | | 0 | 0 |
| | All long-term users (6 m>) | 69 | 18 | 26 | 4 | 22.2 (2.9 to 41.1) |
| | | 134 | 35 | 26 | 13 | 37.1 (21.1 to 53.1) |

*Available population at cup-check in last study quarter.

of the 10%–20% vaginal carriage rate reported in the general population in high-income countries.[29 46] We found no statistical difference in the prevalence of *S. aureus* detected at first introduction and during intervention with cups and pads, with no significant difference between study groups. No cases of mTSS were detected, and TSST-1 was not found among girls using cups with laboratory confirmed *S. aureus* colonisation. No harms were detected although *E. coli* was grown on a third of used cups.

To our knowledge, no other studies have evaluated the safety of menstrual products among schoolgirls in LMIC settings. This study demonstrated the feasibility of such evaluations, and that testing laboratory evaluations can be conducted to ensure rigour. However, as a feasibility study, a number of limitations are evident. For logistic reasons, we were unable to conduct complete sampling for a 'before' and 'after' *S. aureus* prevalence survey. Although sampling at different stages may have introduced bias, we detected no difference in prevalence between girls sampled early during intervention, compared with later or between study groups, and the point prevalence remained within the 'expected' range.[47] Only a small proportion of *S. aureus* positives were identified positive on second testing, leaving few[10] samples for isolation of TSST-1, preventing analysis of risk factors. *S. aureus* was only detected in 20% of girls who had been positive at first screening, when swabbing was repeated. Transience is a recognised phenomenon in vaginal carriage, with higher persistence in nasal than vaginal carriage.[47] Colonisation also varies according to the time of the menstrual cycle due to altered levels of iron, pH, oxygen, carbon dioxide, redox potential and/or osmolarity.[29 48 49] While we found no adverse events among the participants followed, 10% of girls migrated out of the study area. The HDSS system allowed us to visit all homes with no cases of TSS reported from these families. We assessed all health clinic registers, reviewing girls by name, and no TSS cases were found. If a girl was registered under a different name, we would have missed this; however, no cases of mTSS were diagnosed, and our HDSS census identified no study participant had died. We note, however, mTSS is rare and our population studied was small. We used indole test to generate *E. coli* growth[45]; confirmatory tests would more accurately assess contamination risk of *E. coli*,[50] which we assume would have been equally distributed across groups stratified by time. We did not consider evaluating *E. coli* on pads or cloths, which in hindsight may have provided an important comparison.

## Interpretation

Studies in high-income countries have shown menstrual cups to be safe and effective. Post-marketing surveillance of over 100 million soft menstrual cup users, and examination of vaginal pH and microflora, urinalysis, pap smears and colposcopy in 406 subjects using cups for 3 months found no evidence of adverse effects among menstrual cup users.[28] One study examined whether menstrual cups act as a fomite for *S. aureus* or are conducive to contamination with TSST-1; no association was found.[27] Our surveillance did not detect mTSS among participants using menstrual cups or other menstrual care items. mTSS is a rare condition with a risk of 1–16 cases per 100 000 women-years and has a lower mortality than non-menstrual TSS, it may still be life threatening.[51] Menstrual TSS requires vaginal colonisation with a toxin-producing strain of *S. aureus* in the vagina during menstruation, in the absence of a positive antibody (titer of 1:32).[47] TSST-1 is a super antigen, causing an exaggerated release of inflammatory cytokines responsible for the symptoms of clinical disease.[47] It is believed that mTSS develops from a site of colonisation rather than from infection.[29 36] In our study, TSST-1 toxin was detected in 2 of 10 *S. aureus* isolates, a prevalence of 20% in girls with persistent *S. aureus,* similar to the range detected in other studies.[52–54] The two girls, both in the sanitary pad arm with no access to menstrual cups, were healthy and did not exhibit any sign or symptom of mTSS. Other studies have shown the presence of TSST-1 detected among healthy individuals with protective antibody titers (>100).[55] One case of mTSS has been reported in Canada in a woman with Hashimoto thyroiditis, an autoimmune disease, within 10 days of her first cup use.[56] Further surveillance of mTSS among users is, thus, warranted.

*E. coli* growth was detected on a third of cups, with the greatest proportion among girls who had been allocated cups within 6 months. Separate questioning of girls revealed dropping of cups occurred more frequently in early use.[57] Inexperienced girls reported difficulty changing and emptying in school where locks were absent from latrine doors, and conditions were cramped and unlit with the stress of other girls waiting outside to use the latrine.[57] School eligibility required a threshold of 70 pupils or less per latrine, separate toilets for girls and water observed at baseline in this study.[38] We note in a separate paper that hand-washing was reported more commonly among girls using cups than the other groups, suggesting nurse training and caution about cup hygiene was understood by girls.[58] However, other studies show girls' inability to adequately clean themselves after defecation, resulting in vaginal contamination with 10 different micro-organisms including *E. coli*.[59] However, we were unable to also swab girls to assess the presence of vaginal *E. coli*, and thus cannot infer that *E. coli* on cups would be associated with vaginal *E. coli*. Our research supported the hygienic use of all menstrual products, with intense monitoring of all participants over time. We recommend further studies examining the vaginal microbiome among menstrual cup users, include vaginal *E. coli*, as a study reported an association between vaginal *E. coli* and low birth weight.[60] We, thus, offer caution to programmes embarking on menstrual cup or pad distribution to ensure adequate safety procedures, including information, education and communication are provided

to girls, and support of WASH infrastructure in their schools.[61] Similarly, we note provision of hygiene supplies and follow-up of nurses improved the hygienic circumstances for cup use in this LMIC setting.

## CONCLUSIONS

To our knowledge, no studies have evaluated the safety of menstrual products used by schoolgirls in LMIC. In this study, we did not detect harms associated with menstrual cups use among adolescent schoolgirls in this rural African setting. The vaginal colonisation rate of *S. aureus* was within the range of published data, and similarly we observed only a 20% rate of TSST-1 among girls with persistent *S. aureus* colonisation, with no direct association with menstrual cups. Further studies such as large-scale trials and post-marketing surveillance are recommended to verify findings from this feasibility study. Studies are required to further strengthen methodological approaches used in LMICs. Presence of *E. coli* grown on a quarter of sampled cups and higher rates among new users, despite substantive education by study nurses, suggests hygiene education and WASH infrastructures in schools needs to be strengthened,[62 63] and cup provision requires a strong educational component.

**Acknowledgements** We thank the schools, girls, staff and stakeholders for contributing to this study; we thank Mooncups Ltd for providing cups at a discounted price, and the Director of KEMRI for approving the manuscript. The findings and conclusions of this report are those of the authors and do not necessarily represent the official position of the Centers for Disease Control and Prevention.

**Contributors** JJO, LA, CaO, and COO carried out the experiments. EN, KO, JO conducted the field work. JO, EN, PPH and KFL coordinated clinical follow-up. PPH, and CIO performed the statistical analysis. PPH, LM, COO, KA, BF and KFL conceived the study, participated in the design and coordination, and drafted the manuscript. All authors read and approved the final manuscript.

**Funding** This study was funded by the UK Medical Research Council/Department for International Development/Wellcome Trust (Joint Global Health) Trials award scheme (G1100677/1).

**Competing interests** None declared.

**Ethics approval** Kenya Medical Research Institute; Liverpool School of Tropical Medicine.

**Provenance and peer review** Not commissioned; externally peer reviewed.

**Data sharing statement** No additional data are available.

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
