## [Reviewer comments · BMJ Open]

ARTICLE DETAILS

TITLE (PROVISIONAL)	Examining the safety of menstrual cups among rural primary school girls in western Kenya: observational studies nested in a randomized controlled feasibility study
AUTHORS	Juma, Jane; Nyothach, Elizabeth; Laserson, Kayla; Oduor, Clifford; Arita, Lilian; Ouma, Caroline; Oruko, Kelvin; Omoto, Jackton; Mason, Linda; Alexander, Kelly; Fields, Barry; Onyango, Clayton; Phillips-Howard, Penelope

VERSION 1 - REVIEW

REVIEWER	Dr Mags Beksinska Match Research Unit, dept of o&G University of the Witwatersrand, South Africa
REVIEW RETURNED	17-Dec-2016

GENERAL COMMENTS	Thank you this is a very interesting paper and adds to data that young women in low resourced settings can use a menstrual cup with training. It is a well written and informative article and i recommend it be published, well done. I have only a few comments:- The introduction or perhaps the discussion could mention or compare with similar outcomes for tampons. although I note in the introduction there are references mentioned which probably have some data on tampons there could be something added to the text? e.g Tampons are linked to menstrual toxic shock syndrome (TSS) in women of reproductive age. Surveillance data for the period 1979 to 1996 indicates when 5,296 cases in the US were reported with super absorbant tampons being identified as causing more cases. Also could say something about the material and the fact that it can be compared to other vaginal products as the menstrual cup is made of the same material as contraceptive barriers such as diaphragms and cervical cups. Silicone barriers are often left in overnight or fitted several hours before use. Their safety has been documented over many years. The menstrual cup can be compared to these devices. The washing instructions for both devices are similar and I do remember hearing of TSS with a latex diaphragm during menses but not otherwise with silicone cervical cups or diaphragms. Results the results should start with a description of the participants? have they ever used tampons before? ever used pads etc how long had they been menstruating (although we know they had had at least 3 cycles- their average age. we know they were between 14-16 but things like mean age, parity? ever sexually active etc would be useful to frame the population at the start of the results. There are many references maybe too many?
--

REVIEWER	Supriya D. Mehta
-----------------	------------------

	University of Illinois at Chicago United States of America
REVIEW RETURNED	19-Dec-2016

GENERAL COMMENTS	Review safety of menstrual cup paper This manuscript summarizes the results of a study of safety endpoints related to menstrual cup use among adolescent girls in rural Kenya. The authors demonstrate no difference in S aureus detection among cup users compared to pad users and controls. Overall, the manuscript is generally well-written, though some minor clarifications are requested. The manuscript provides necessary information about the safety of menstrual cups for young girls with limited or poor WASH facilities, demonstrating the feasibility and safety for this cost-effective approach to menstrual management in this setting. Abstract 1. Participants: It appears part of the sentence has been dropped. 2. Results: I suggest the authors should move sample size to the first sentence, as well as duration of follow-up observation. "Among 604 eligible girls tested, no adverse event or TSS was detected over median 9 months follow-up. S. aureus prevalence was 10.8%, with no significant difference..." Introduction 3. Paragraph 2, sentence 1 (lines 29-31): Revise first sentence as there is a disjoint between "association" clause and outcome. E.g., "...found no association between the frequency of ...infections and menstrual cup use...." 4. Paragraph 2, sentence 3 (lines 30-42). Reads a bit like the cups do not absorb the vaginal epithelium. Suggest changing "and do not absorb" to "collect blood, are non-absorptive, and do not disrupt..." Methods 5. Ms stuy, page 5. The authors state that a sample of 185 girls per arm would be selected from a total population of 3,165 girls. Could the authors please clarify – is the 3,165 girls the total eligible among the 30 schools? Could the authors please add a sentence explaining how selection and sampling were conducted? Could the authors please include description of the duration of study follow-up here? 6. Menstrual products and hygiene, page 6, para 1. Could the authors please clarify: They state that girls in the sanitary pad arm were each given 2 pack (total 16 pads) monthly. Could they request and receive more if the quantity was insufficient? How were girls in the sanitary pad arm instructed to dispose of used pads? 7. Vaginal swabbing, pp. 7-8. Sentence. "This activity involved taking a vaginal swab (approximately 3 to 4 inches into the vagina) in the school bathroom or private room". Perhaps this should be two sentences, or rephrase. "The self-collected vaginal swab was collected by girls in a school bathroom or private room, and obtained
---

by inserting the swab approximately 3-4 inches into the vagina.”

Some questions:

- 1) 3-4 inches seems quite far?
- 2) were they instructed to twirl it or leave it in for any period of time?
- 3) What type of swab was used?
- 4) How frequently were specimens shipped to KEMRI laboratories (daily, weekly) and how were they stored interim?

8. Menstrual cup screening for E. coli contamination, page 9, para 1. The first sentence is very confusing. Among all girls, a 25% random sample was selected, stratified by...., excluding girls..? Did any girls refuse to swab their existing cup?

9. No statistical analysis methods are described but p-values are reported.

Results

10. Results, page 9. Could the authors please add a few sentences describing the enrolled study population? The information on the menstrual cup loss or damage might read better before outcome reporting. A table 1 with N by arm and distribution of characteristics, median duration of follow-up, any loss to follow-up, prevalence of S. aureus would be helpful.

11. Menstrual cup loss or damage, pp.10-11. Authors note “Of 225 girls provided cups, 29 were LFU...” and then “Of 195 retained girls,” Could authors please clarify whether 30 were LRU or whether 196 were retained? Please insert a colon to clarify the list follows, “reported: cup loss...”

12. E. coli growth on menstrual cups, p 11, para 1. The authors note that 134 girls remained in the last quarter of the study. This is somewhat confusing because the expected number is 196 from the preceding paragraph. Could the authors please explain whether the difference between 134 and 196 is due to LFU or completion of study protocol? Cup prevalence differed by duration of use (new user vs. established); curious - was there any difference by age (admittedly, narrow age range)? Any differences by school, handwashing, or WASH characteristics?

Tables and Figures

Recommend add a Table 1 of distribution of characteristics and findings by study arm.

Figure 3. For this figure, the y-axis range (0-100%) should be lowered to 25% upper limit to be able to visualize small differences between groups. Alternatively, a table with prevalence by arm as columns and time as rows might be more informative. Is the trend for increasing prevalence of S. aureus for cups, pads, controls at 1 month significant?

Discussion

13. Page 11, Sentence 2. Authors note “We found no statistical difference in the prevalence of S. aureus detected before and during intervention..” This is confusing because the Methods do not indicate collection of a self-collected vaginal swab prior to intervention, nor are the results for prevalence before intervention reported, and the

	subsequent paragraph states “For logistic reasons we were unable to conduct complete sampling for a ‘before’ and ‘after’ S aureus prevalence survey” 14. Page 11, Line 52. Authors report “No harms were detected although E. coli was detected on a third of used cups. Could the authors please discuss what they think this finding reflects?” 15. Page 12, Line 17. Suggest changing ‘normal’ to “expected”. 16. Page 13, Line 49/50 “E. coli was generated on a quarter of cups...” recommend change to “E. coli was detected”. Also, could the authors please clarify as Page 11, first paragraph of Discussion, last sentence on page 11 (line 52) notes “...E. coli was grown on a third of used cups.”
--	---

VERSION 1 – AUTHOR RESPONSE

Reviewer: 1

Thank you this is a very interesting paper and adds to data that young women in low resourced settings can use a menstrual cup with training. It is a well written and informative article and i recommend it be published, well done. I have only a few comments:-

Reviewer 1: Comment 1, Introduction

The introduction or perhaps the discussion could mention or compare with similar outcomes for tampons. although I note in the introduction there are references mentioned which probably have some data on tampons there could be something added to the text? e.g Tampons are linked to menstrual toxic shock syndrome (TSS) in women of reproductive age. Surveillance data for the period 1979 to 1996 indicates when 5,296 cases in the US were reported with super absorbent tampons being identified as causing more cases.

Response: We are grateful for the opportunity to expand the text on this, and have included this, as suggested in the introduction, adding Hajjeh et al 1999 to the references, as follows:

Introduction, p4: Tampons are linked to mTSS in women of reproductive age. Surveillance data for the period 1979 to 1996 indicates 5,296 cases were reported in women in the USA using highly absorbent tampons (30). The tampons were found to have been associated with vaginal micro-trauma arising from the high absorbency (27, 30-33).

Reviewer 1: Comment 2, Introduction

Also could say something about the material and the fact that it can be compared to other vaginal products as the menstrual cup is made of the same material as contraceptive barriers such as diaphragms and cervical cups. Silicone barriers are often left in overnight or fitted several hours before use. Their safety has been documented over many years. The menstrual cup can be compared to these devices. The washing instructions for both devices are similar and I do remember hearing of TSS with a latex diaphragm during menses but not otherwise with silicone cervical cups or diaphragms.

Response: We agree and have added the following:

Introduction, p4: Further, among women using female barrier methods, which similarly uses medical grade silicone or latex products, is very low (~2.25 cases per 100 000 users per year) (34).

Reviewer 1: Comment 2, Results

The results should start with a description of the participants? have they ever used tampons before?

ever used pads etc how long had they been menstruating (although we know they had had at least 3 cycles- their average age. we know they were between 14-16 but things like mean age, parity? ever sexually active etc would be useful to frame the population at the start of the results.

Response: We have now added the participant description, as Table 1.

Reviewer 1: Comment 3, References

There are many references maybe too many?

Response: Thanks for this thought. We feel it is a neglected field of study, with few studies published, particularly with a lack of menstrual safety studies in LMIC, including as the reviewer noted, inadequate comparison with other materials used vaginally. We are keen thus to keep as many references as possible for interested readers, and to inform researchers commencing work on MHM who have no reproductive health background. We needed to add a few extra references regarding other materials, and so have cut back some others that duplicate methods or findings, leaving a similar overall number.

Reviewer 2:

This manuscript summarizes the results of a study of safety endpoints related to menstrual cup use among adolescent girls in rural Kenya. The authors demonstrate no difference in *S aureus* detection among cup users compared to pad users and controls. Overall, the manuscript is generally well-written, though some minor clarifications are requested. The manuscript provides necessary information about the safety of menstrual cups for young girls with limited or poor WASH facilities, demonstrating the feasibility and safety for this cost-effective approach to menstrual management in this setting.

Reviewer 2: Comments, Abstract

1. Participants: It appears part of the sentence has been dropped.

Response: Thank you for noting - the word 'study' was missing.

2. Results: I suggest the authors should move sample size to the first sentence, as well as duration of follow-up observation. "Among 604 eligible girls tested, no adverse event or TSS was detected over median 9 months follow-up. *S. aureus* prevalence was 10.8%, with no significant difference..."

Response: We have amended as suggested (median is 10.9m).

Reviewer 2: Comments, Introduction

3. Paragraph 2, sentence 1 (lines 29-31): Revise first sentence as there is a disjoint between "association" clause and outcome. E.g., "...found no association between the frequency of ...infections and menstrual cup use..."

Response: We have edited this, as follows:

Introduction, p4: While menstrual cups have not been associated with an increased risk of reproductive tract and urogenital infections in women in high income countries....

4. Paragraph 2, sentence 3 (lines 30-42). Reads a bit like the cups do not absorb the vaginal

epithelium. Suggest changing “and do not absorb” to “collect blood, are non-absorptive, and do not disrupt...”

Response: Thanks, we edited as suggested.

Reviewer 2: Comments, Methods

5. Ms Study, page 5. The authors state that a sample of 185 girls per arm would be selected from a total population of 3,165 girls. Could the authors please clarify – is the 3,165 girls the total eligible among the 30 schools? Could the authors please add a sentence explaining how selection and sampling were conducted? Could the authors please include description of the duration of study follow-up here?

Response: 3,165 girls was an estimate of the total population of target aged girls in the Gem study area, not in the 30 schools under study. We have removed this demographic to reduce misunderstanding, as all girls in the 30 study schools, aged 14-16 years, were eligible, with no selection of a sub-sample. We have added a sentence on study enrolment and follow-up, as below, and also reference the published BMJ Open paper on the main pilot study. We also present a Flow Diagram of the study population.

p6: Girls were enrolled from August 15, 2012 to August 27, 2013 and followed until November 21, 2013, with a median (IQR) follow-up time of 10.9 (6.1-12.5) months. Further details of the overall study methods are published elsewhere (39).

6. Menstrual products and hygiene, page 6, para 1. Could the authors please clarify: They state that girls in the sanitary pad arm were each given 2 pack (total 16 pads) monthly. Could they request and receive more if the quantity was insufficient? How were girls in the sanitary pad arm instructed to dispose of used pads?

Response: We estimated 2 packs of 16 pads were sufficient (e.g. 4 pads/day with an average menses of 3.8 days). Girls could request more if insufficient, and spoke with the nurse about heavy menses. The 10 girls reporting particularly heavy periods were referred to the obstetrician / gynaecologist for assessment with nothing abnormal detected. Participants were trained to dispose of their pads and pad wrappers in the designated waste bin, and not to drop on floors or in the latrine pit.

7. Vaginal swabbing, pp. 7-8. Sentence. “This activity involved taking a vaginal swab (approximately 3 to 4 inches into the vagina) in the school bathroom or private room”. Perhaps this should be two sentences, or rephrase. “The self-collected vaginal swab was collected by girls in a school bathroom or private room, and obtained by inserting the swab approximately 3-4 inches into the vagina.” Some questions: 1) 3-4 inches seems quite far? 2) were they instructed to twirl it or leave it in for any period of time? 3) What type of swab was used? 4) How frequently were specimens shipped to KEMRI laboratories (daily, weekly) and how were they stored interim?

Response: Text has been amended to the description used by the nurses for training the participants, as follows:

p8: The self-collected vaginal swab (BBL Culture swab, COPLAN for Becton Dickinson) was conducted by girls in a school bathroom or private room. Participants were taught to insert the swab into the endocervical canal and stop when the tip was no longer visible. As instructed, the girls would then rotate the swab 3-5 times inside the vagina, withdraw it but avoid contact with vaginal surfaces, and put the swab in the tube (containing Ames transport media with agar), break the handle of the swab and close the tube tightly.

(4) Swabs were transported daily; transport followed the SOP, the text included in the manuscript is

as follows: P8 - Swabs were transported on the same day, within 6 hours of collection, to the KEMRI laboratory, and stored at -70oC to -80oC before testing.

8. Menstrual cup screening for E. coli contamination, page 9, para 1. The first sentence is very confusing. Among all girls, a 25% random sample was selected, stratified by...., excluding girls..? Did any girls refuse to swab their existing cup?

Response: There were no refusals. We have amended this section as follows – see similarly Reviewer 2, Comments 11-12.

p12: Of 188 girls provided cups and followed to outcome, 21 dropped-out. Of the 167 non-dropouts, a sampling frame was obtained from the nurse’s follow-up survey database in the last study quarter. This comprised 134 girls surveyed, with 33 girls missed due to non-attendance. From this sample of 134 girls, a random selection of 1 in 4 cups (35, 26%) were randomly selected. No girls refused.

9. No statistical analysis methods are described but p-values are reported.

Response: We have now included a section on data analysis (p10).

Reviewer 2: Comments, Results

10. Results, page 9. Could the authors please add a few sentences describing the enrolled study population? The information on the menstrual cup loss or damage might read better before outcome reporting. A table 1 with N by arm and distribution of characteristics, median duration of follow-up, any loss to follow-up, prevalence of S. aureus would be helpful.

Response: Thank you, as also requested by Reviewer 1, in Comment 2. We have now included table 1 population characteristics, including median follow-up. We now also present a flow diagram of the loss of the population during the study, and total sampled for S aureus. Please see response Reviewer2, Tables and Figures on S aureus; with table 2 providing prevalence values for S aureus.

11. Menstrual cup loss or damage, pp.10-11. Authors note “Of 225 girls provided cups, 29 were LFU...” and then “Of 195 retained girls,” Could authors please clarify whether 30 were LRU or whether 196 were retained? Please insert a colon to clarify the list follows, “reported: cup loss...”

Response: We note the confusion, as the initial submission had documented as the denominator for cup loss/damage to all girls, including those migrating. To simplify, we restrict this now to the 188 girls in the cup group in the total 644 population followed to outcome (as per the flow diagram).

p12: Of 188 girls followed to outcome, 14 (7%) girls required replacement cups; due to cup loss (3 participants, including one girl twice), damage (2; one burned cup when boiling, one cup eaten by rats), too small (3, replaced with size A due to leaking), or dropped inside the latrine (6 cups).

12. E. coli growth on menstrual cups, p 11, para 1. The authors note that 134 girls remained in the last quarter of the study. This is somewhat confusing because the expected number is 196 from the preceding paragraph. Could the authors please explain whether the difference between 134 and 196 is due to LFU or completion of study protocol? Cup prevalence differed by duration of use (new user vs. established); curious - was there any difference by age (admittedly, narrow age range)? Any differences by school, handwashing, or WASH characteristics?

Response: We have clarified the sampling frame as below, noting as above (Q11), limiting the sample to the 188 girls followed:

p12: Of 188 girls provided cups and followed to outcome, 21 dropped-out. Of the 167 non-dropouts, a sampling frame was obtained from the nurse's follow-up survey database in the last study quarter. This comprised 134 girls surveyed, with 33 girls missed due to non-attendance. From this sample of 134 girls, a random selection of 1 in 4 cups (35, 26%) were randomly selected. No girls refused.

Response: We examined girls' age, socio-economic status, reported ever used pads, age at menarche, duration of bleeding, and if periods were reported to be heavy as characteristics – of these, heavy periods were associated in bivariate analysis (among girls reporting heavy periods 61.5% E.coli on cups, compared with 22.7% of girls stating they did not have heavy periods; $p=0.022$). The number of cups examined was too small to perform multivariate analysis.

p13: Examination of E coli growth by girls' age, socio-economic status, reported ever used pads, age at menarche, duration of bleeding, and if periods were reported to be heavy, only found an association with heavy periods; 61.5% of girls reporting heavy periods had E.coli on cups, compared with 22.7% of those stating they did not have heavy periods($p=0.022$).

Reviewer 2: Comments, Tables and Figures

Recommend add a Table 1 of distribution of characteristics and findings by study arm.

Response: Table 1 has been added as suggested.

Figure 3. For this figure, the y-axis range (0-100%) should be lowered to 25% upper limit to be able to visualize small differences between groups. Alternatively, a table with prevalence by arm as columns and time as rows might be more informative. Is the trend for increasing prevalence of S. aureus for cups, pads, controls at 1 month significant?

Response: Thank you, this is now presented as a table (table 2); with prevalence values and significance values, showing no difference statistically between groups, or between early and later sampling.

Reviewer 2: Comments, Discussion

13. Page 11, Sentence 2. Authors note "We found no statistical difference in the prevalence of S. aureus detected before and during intervention.." This is confusing because the Methods do not indicate collection of a self-collected vaginal swab prior to intervention, nor are the results for prevalence before intervention reported, and the subsequent paragraph states "For logistic reasons we were unable to conduct complete sampling for a 'before' and 'after' S aureus prevalence survey"

Response: We have edited sentence 2, as we agree use of the term 'before' is confusing and inaccurate. Similarly, we edited the section on S aureus prevalence survey; these now read as follows:

p11: There was no also significant difference in prevalence between early intervention and during use, overall or within groups (table 2).

p13: We found no statistical difference in the prevalence of S aureus detected at first introduction and during intervention.

14. Page 11, Line 52. Authors report "No harms were detected although E. coli was detected on a third of used cups. Could the authors please discuss what they think this finding reflects?"

Response: Thank you in the final paragraph on p15 we discuss that girls reported a higher frequency of dropping cup in early use, as girls were inexperienced and had difficulty emptying and reinserting the cup without dropping on the latrine floor. A Turkish study noted that vaginal e.coli occurs after girls do not adequately clean themselves after defecation, which may also have occurred more frequently in early users. We have added the following to this discussion:

P15: However, we were unable to also swab girls to assess presence of vaginal E.coli, and thus cannot infer that E.coli on cups would be associated with vaginal E.coli. We recommend further studies examining the vaginal microbiome among menstrual cup users, include vaginal E.coli, as a study reported an association between vaginal E.coli and low birth weight (ref).

15. Page 12, Line 17. Suggest changing 'normal' to "expected".

Response: This has been changed (p14).

16. Page 13, Line 49/50 "E. coli was generated on a quarter of cups..." recommend change to "E. coli was detected". Also, could the authors please clarify as Page 11, first paragraph of Discussion, last sentence on page 11 (line 52) notes "...E. coli was grown on a third of used cups."

Response: The word 'generated' was changed to 'detected' (p15). The reviewer found an error – both sentences should state 'a third', this is amended accordingly.

VERSION 2 – REVIEW

REVIEWER	Mags Beksinska MRU MatCH Research Unit, University of the Witwatersrand, South Africa
REVIEW RETURNED	05-Feb-2017

GENERAL COMMENTS	I have reviewed this paper before and noted all my comments were adressed thank you. Just a thought? the ecoli on the cups can some of this growth be attributed to possible length of time cup sealed in the zip lock bag mentioned before analysis. Noting also we did some work on how hands/ handling/ drying with towels which themselves may be dirty can transfer E coli onto barrier methods. this could be mentioned in the discussion? but not essential. thanks a very good paper.
---

REVIEWER	Supriya D. Mehta Associate Professor of Epidemiology School of Public Health University of Illinois at Chicago Chicago, IL
REVIEW RETURNED	26-Jan-2017

GENERAL COMMENTS	Concerns addressed
--------------------